# Splenic Artery Aneurysms, a Rare Complication of Type 1 Gaucher Disease: Report of Five Cases

**DOI:** 10.3390/jcm8020219

**Published:** 2019-02-08

**Authors:** Christine Serratrice, Timothy M. Cox, Vanessa Leguy-Seguin, Elizabeth Morris, Karima Yousfi, Olivier Monnet, Annie Sibert, Wassim Allaham, Nadia Belmatoug

**Affiliations:** 1Department of Internal Medicine and Rehabilitation, University Hospital of Geneva, Hôpital des Trois Chêne, 1226 Thonex, Switzerland; 2Division of Internal Medicine of the Aged, University Hospital of Geneva, 1205 Geneva, Switzerland; 3Department of Internal Medicine, Saint Joseph Hospital, 13008 Marseille, France; 4Department of Medicine, University of Cambridge, Cambridge CB2 0QQ, UK; tmc12@medschl.cam.ac.uk; 5Lysosomal Disorders Unit, Addenbrooke’s Hospital, Cambridge CB2 0QQ, UK; liz.morris@addenbrookes.nhs.uk; 6Department of Internal Medicine and Clinical Immunology, University Hospital of Dijon, 21000 Dijon, France; vanessa.leguy-seguin@chu-dijon.fr; 7Reference Centre of Lysosomal Diseases, University Hospital Beaujon, 92110 Clichy, France; karima.yousfi@aphp.fr (K.Y.); nadia.belmatoug@aphp.fr (N.B.); 8Department of Radiology, Saint Joseph Hospital, 13008 Marseille, France; omonnet@hopital-saint-joseph.fr; 9Department of Radiology, University Hospital Beaujon, 92110 Clichy, France; annie.sibert@aphp.fr (A.S.); wassim.allaham@aphp.fr (W.A.)

**Keywords:** Gaucher disease, lysosomal storage disorder, splenic arterial aneuryms, enzyme replacement therapy

## Abstract

Type 1 Gaucher disease is a rare genetic lysosomal disorder due to acid betaglucosidase deficiency. The main features are thrombocytopenia, anemia, hepatosplenomegaly and complex skeletal disease. Complications include pulmonary hypertension, cirrhosis and splenic infarction; comorbidities, such as autoimmune phenomena, B-cell malignancies and Parkinson disease also occur. Visceral aneurysms have been only rarely noted in Gaucher disease. We report the retrospective data from patients with Gaucher disease type 1 and splenic arterial aneurysm. We describe the different outcomes of a giant splenic arterial aneurysm in five patients with type 1 Gaucher disease and discuss the main possible pathophysiological explanations. Aneurysms of the splenic artery are rare in Gaucher disease but are probably greatly under-reported.

## 1. Introduction

Gaucher disease (GD) is a rare inborn error of lysosomal metabolism characterized by lysosomal storage of the sphingolipid, β-glucosylceramide. GD is due to the deficiency of a lysosomal enzyme, acid beta-glucosidase (or glucocerebrosidase) [1] or in rare cases of its activator, saposin C [2]. The disease incidence is around 1/50,000 births in the general population and 1/800 births in Ashkenazi Jewish population [3,4]. GD diagnosis is confirmed by showing low acid-β-glucosidase activity (<30%) in peripheral leucocytes [5]. Mutations in the gene, *GBA1*, encoding β-glucosylceramidase disable the enzyme and β-glucosylceramide accumulates within macrophages [6].

Three different subtypes of GD are recognized: type 1 GD (GD1) is characterized by thrombocytopenia, anemia, an enlarged spleen and liver—as well as bone complications (Erlenmeyer flask deformity, osteoporosis, lytic lesions, pathological and vertebral fractures, bone infarcts and avascular necrosis leading to degenerative arthropathy) [7]. GD1 represents 90% of cases in Western countries. Type 2 GD is the acute infantile neuronopathic form, which progresses rapidly and causes death within the first year or two of life [8]. Type 3, or chronic neuronopathic GD, is a “catch all” encompassing patients who survive infancy but have highly variable but progressive neurological involvement. In reality these phenotypes overlap, and GD is now viewed as a disease continuum [9].

GD1 may be complicated by pulmonary hypertension, cirrhosis, splenic infarction or rupture, several cancers and/or Parkinson disease. Splenic artery aneurysms (SAAs) have been little recognized. The splenic artery is defined as aneurysmal when a focal dilatation is observed and its diameter greater than 50% of the normal vessel diameter. SAAs account for more than half of all visceral aneurysms and are the third most frequent intra-abdominal aneurysms [10]. True SAAs involve all layers of the arterial wall, each of which is intact but attenuated. Pseudoaneurysms are the result of a tear in the vessel wall with dissection of the intima: these may be the consequence of trauma or inflammation of the splenic artery. Small SAAs (≤2 cm) are usually asymptomatic, but giant SAAs (≥5 cm) are more likely to cause symptoms and can result in life-threatening complications. Hypertension, atherosclerosis, cirrhosis, portal hypertension, liver transplantation, female sex (with pregnancy and multiparity) and connective tissue disorder such as Marfan or Ehlers-Danlos syndromes are recognized associations.

Although splenomegaly in GD has been associated with SAAs, a bibliographic review identified only two reports of this complication [11]. We report the retrospective data from five patients with GD1 and SAA. All the patients have abdominal imagery (ultrasonography, CT scan or MRI) according to the recommendations for GD1. All the patients gave their informed consent. Ethics approval was not required for these case reports.

Three men and two women with GD1 were diagnosed with SAA. Patient characteristics are set out in Table 1. 

The patients had splenomegaly and moderate thrombocytopenia at the diagnosis of GD but in one patient detailed records of diagnosis were not found. Four out of five had bone manifestations at the diagnosis of GD (Erlenmeyer flask deformity, bone infarcts and bone marrow infiltration). Three patients had received enzyme replacement therapy (ERT) before the diagnosis of SAA. One patient (Patient 1) was treated between 1993 and 1999 but for personal reasons stopped the therapy; in this patient SAA was identified post mortem after an acute and catastrophic intraabdominal hemorrhage. One patient stopped ERT after three years. Two patients started the treatment after the SAA was discovered.

ERT improved the platelet count and splenomegaly in all the patients so treated. SAA diameter was 40 mm for patient 2, 30 mm for patient 3 (Figure 1), 29 mm for patient 4 and 24 mm for patient 5. SAA size increased for patient 3 (+6 mm between 2009 and 2018 despite ERT) and was stable for two patients (patients 4 and 5). 

Patient 1 died rapidly after bleeding, which occurred while she was awaiting an underground train. Coroner’s post-mortem was decisive: there was massive blood loss into the abdomen with a 10 × 15 cm collection in the splenic bed. The spleen itself was firm and intact: the only abnormality was a dilated thin-walled arterial aneurysm at the hilum of the spleen indicating a true SAA. Patient 2 presented a set of four SAA and has been splenectomized. Patient 4 was successfully embolized. Patient 3 refused embolization or partial splenectomy but is now carefully and regularly monitored, as patient 5.

## 2. Discussion

The prevalence of SAA associated with GD in this study is 2% (5 patients out of 252 GD1 patients followed in these 4 centers) whereas the prevalence in the general population is extremely variable between 1% [12] to 10.4% in necropsy series [13]. However, this prevalence must be considered with cautiousness as in GD1, such an aneurysm can be underdiagnosed with abdominal ultrasonography. Indeed, a total splenic artery exploration is quite difficult due to their deep course and the gut superpositions, especially if the aneurysm is not in the hilum. Moreover, in GD, due to the splenomegaly it is frequent to observe a tortuous splenic artery with loops making the diagnosis of SAA difficult. Similarly, SAAs may be underdiagnosed in the general population because they are often asymptomatic [14]. To our knowledge, only two cases of splenic arterial aneurysm associated with GD1 have been reported in the literature in 1989 [11]. Recently Reynolds et al. described a patient with multiple intracerebral aneurysms and GD1 [15]. Nasu et al. described an annulo-aortic ectasia with type II dissecting aneurysm in a Gaucher patient [16]. However, as illustrated here, SAAs are not unusual during the course of this disease. 

Most of those reported here were asymptomatic or initially asymptomatic, except for those occurring in the two women: (1) the first, who was a patient found to have the condition at post mortem examination after rupture, had a thankfully brief symptomatic period as fatal hemorrhagic shock supervened; (2) the other woman, complained of worsening pain in the left hypochondrium and had mild local tenderness. While this patient had frequently complained of mid pain in the region over many years and had been known to have at least one aneurysm, the changed character of the symptoms and their provocation by external jolting movements, led to magnetic resonance (MR) angiography at which several large aneurysms were detected (the largest at >4 cm diameter). By this time, a vascular surgical opinion was sought urgently, and curative splenectomy rather than embolization was successfully carried out with good symptom relief. These events are consistent with what is known for other SAAs: True splenic arterial aneurysms are asymptomatic until they expand to merit classification as giant or near-giant; they are often diagnosed by means of advanced imaging techniques [17]. We contend that on account of their potential for expansion and rupture SAAs should be searched for actively in patients with Gaucher disease.

In those patients with optimal responses to sustained enzyme therapy as illustrated by patients 4 and 5, aneurysmal size appears not to have changed–albeit with follow-up of only two years.

As to causation, we suggest that the most plausible pathophysiological explanation may be the effects of greatly increased arterial flow attributable to often marked and sustained splenomegaly in this chronic genetic disease. As in pregnancy, or other cases in which massive splenomegaly is sustained, or in the presence of portal venous hypertension, arterial blood flow is likely to become turbulent, exerting strong sheer forces on the inner wall of the artery, as it traverses the constitutively tortuous splenic vessel. Of note three out of five of the patients reported here had no symptoms or signs indicating portal hypertension, and in two cases, SAA was diagnosed despite salutary spleen volume response to ERT. An alternative, or perhaps additional mechanism, would be a change in arterial wall structure and function due to infiltration with pathological macrophages known as Gaucher cells, as has been reported in some patients within pulmonary hypertension that complicates rare cases of GD [18]. So far however, in two of the patients histological material was available and this did not confirm such pathology in the arterial wall.

All these patients were heterozygous for the most usual mutation in GD1, the N370S mutation. However, we cannot conclude that SAA were related to this mutation.

While we are not able to speculate on the impact of targeted Gaucher-specific therapies for GD1 on the behavior of the aneurysms but, it is notable that three patients out of five had no ERT at the time the aneurysm was detected. On first principles, we can suppose that reducing the volume of the spleen would be beneficial so reducing flow, sheer forces and pressure effects in the artery. However, for patient 3 aneurysm diameter increased despite a reduction in spleen volume. The other male patient was successfully treated by splenic artery embolization.

## 3. Conclusions

Aneurysms of the splenic artery are rare in Gaucher disease but are probably greatly under-reported. However, since imaging, including abdominal ultrasonography and/or MR is conducted periodically in many patients receiving specialist care for their Gaucher disease, we contend that more scrupulous examinations and monitoring of aneurysmal behavior is indicated. This will improve knowledge of the course of the condition and is likely to prevent adverse outcomes and pain from intrabdominal pressure effects or even arterial rupture. Despite the absence of consensus on the therapeutic management of these aneurysms and particularly those associated with GD, spectacular progress in radiological techniques has been made—alongside improving treatment choices in Gaucher disease. A coordinated initiative in the field may facilitate risk stratification and determine whether aneurysms arising in a particular context or disease stage will expand and whether specific therapy can mitigate this hazard.

## Figures and Tables

**Figure 1 jcm-08-00219-f001:**
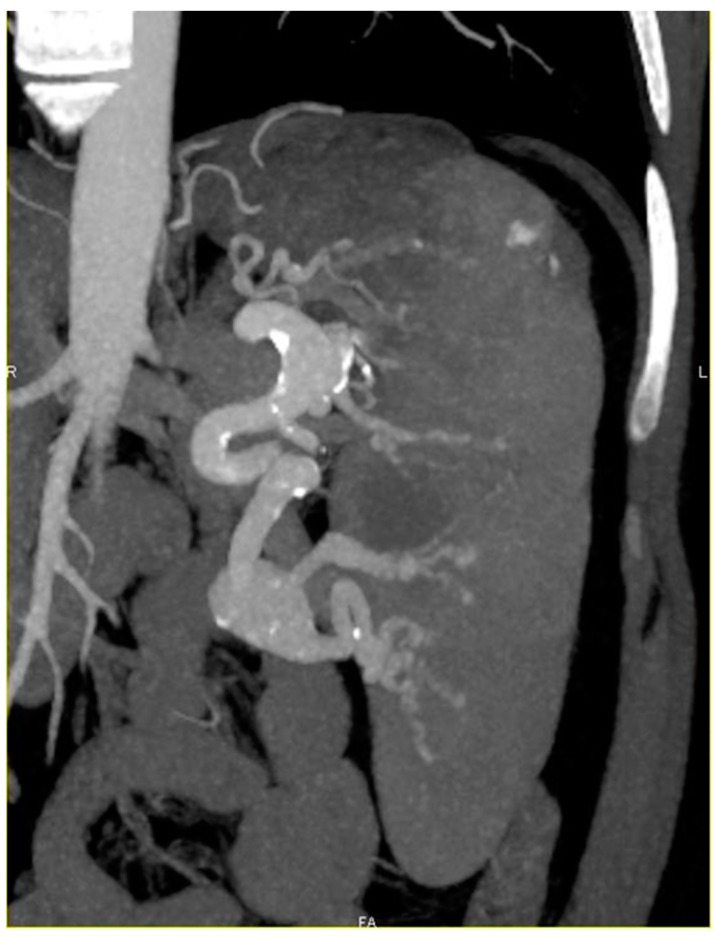
Computerized tomography (CT) scan with coronal reconstruction showing splenomegaly, hilar giant splenic artery aneurysm and intra parenchymatous dysplastic splenic artery with several micro aneurysms.

**Table 1 jcm-08-00219-t001:** Characteristics of patients with aneurysms of the splenic artery.

Patient	Sex/Age	Age at Diagnosis of GD	Genotype	Treatment	Age at Diagnosis of SAA	Evolution
1	F, NA	30	N370S/N370S	Intermittent ERT 10 U/kg eow from 44 to 52-year-old	63	Sudden collapse and death aged 63 years
2	F, 59	19	N370S/84GG	ERT 16 U/kg eow	49	Complex set of four SAA
3	M, 50	10	N370S/N370S	ERT 60 U/kg eow at 40-year-old	40	Aneurism size increase
4	M, 62	38	N370S/D218A	ERT 75 U/kg every three weeks from 45 to 48-year-old	58	Embolization two years later
5	M, 31	29	N370S/K196E	ERT 60 U/kg/eow at 29-year-old then eliglustat at 30-year-old	29	Aneurism size stabilization

F, female; M, male; eow: every other week, ERT: enzyme replacement therapy, SAA: splenic artery aneurysm.

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
