# Peer review of "Splenic Artery Aneurysms, a Rare Complication of Type 1 Gaucher Disease: Report of Five Cases"

_jcm, 2019, doi:10.3390/jcm8020219_

Round 1
Reviewer 1 Report
Serratrice et al. describe several cases of splenic arterial aneurysm in patients with type 1 Gaucher disease (GD). The frequency in GD patients does not seem to be higher than in the general population, but the authors suggest that it might be underdiagnosed in GD. This short MS is well written and provides information that might be useful for the field.
Minor points:
Line 36: Reference 2 is not correct for the point the authors mentioned. Instead, either Christomanou, H., Chabas, A., Pampols, T. and Guardiola, A. (1989) Activator protein
deficient Gaucher disease. Klin. Wochenschr. 67, 999–1003, and or Schnabel, D., Schroder, M. and Sandhoff, K. (1991) Mutation in the sphingolipid activator
protein 2 in a patient with a variant of Gaucher disease. FEBS Lett. 284, 57–59, should be referenced (as mentioned within reference 2).
Line 43: add a space before “Type 3”.
Line 103: add “a” before “patient”
Author Response
Dear reviewer
Thank you for your comments:
Line 36: Reference 2 is not correct for the point the authors mentioned. Instead, either Christomanou, H., Chabas, A., Pampols, T. and Guardiola, A. (1989) Activator protein
deficient Gaucher disease. Klin. Wochenschr. 67, 999–1003, and or Schnabel, D., Schroder, M. and Sandhoff, K. (1991) Mutation in the sphingolipid activator
protein 2 in a patient with a variant of Gaucher disease. FEBS Lett. 284, 57–59, should be referenced (as mentioned within reference 2).
Thank you very much, the reference has been changed
Line 43: add a space before “Type 3”.
The space has been added
Line 103: add “a” before “patient”
The “a” has been added
Reviewer 2 Report
Serratrice et al. present 5 clinical cases of splenic artery aneurysms in patients with type 1 Gaucher disease. Previously, this complication had only been reported once. Therefore, it is of great interest to the medical community to raise the awareness of this complication in Gaucher disease patients.
Questions for the authors:
- In the first line of the “Discussion” it cannot be stated that the prevalence was 2% in the study without explaining that splenic artery aneurysms were searched in the studied Gaucher disease population. What is the number of patients studied?
- Were splenic artery aneurysms related to any specific mutation?
Author Response
Dear reviewer
Thank you for your comments
In the first line of the “Discussion” it cannot be stated that the prevalence was 2% in the study without explaining that splenic artery aneurysms were searched in the studied Gaucher disease population. What is the number of patients studied?
Thank you for this comment. I have added the number of patients followed in our centres
"5 patients out of 252 GD1 patients followed in these 4 centres"
Were splenic artery aneurysms related to any specific mutation?
I have added in the discussion:
“All these patients were heterozygous for the most usual mutation in GD1, the N370S mutation. However we can’t conclude that splenic artery aneurysms were related to this mutation."